# Relationship between Visual Perception and Microstructural Change of the Superior Longitudinal Fasciculus in Patients with Brain Injury in the Right Hemisphere: A Preliminary Diffusion Tensor Tractography Study

**DOI:** 10.3390/diagnostics10090641

**Published:** 2020-08-27

**Authors:** Su-Hong Kim, Hyeong-Eun Jeon, Chan-Hyuk Park

**Affiliations:** Department of Physical & Rehabilitation Medicine, Inha University School of medicine, Inha University Hospital, Incheon 22332, Korea; suhong1207@gmail.com (S.-H.K.); optaum@naver.com (H.-E.J.)

**Keywords:** visual perception, neglect, superior longitudinal fasciculus, diffusion tensor imaging, lateralization

## Abstract

Right hemisphere brain damage often results in visual-spatial deficits. Because various microstructural changes of the superior longitudinal fasciculus (SLF) after a stroke in the right hemisphere affect visual perception, including neglect, the present study investigates the relationship between both microstructural change and lateralization of SLF and visual perception, using diffusion tensor imaging (DTI) in patients with lesions in the right hemisphere. Eight patients with strokes (five patients with intracranial hemorrhage, and three patients with infarction; mean age of 52.5 years) and 16 mean-age-matched healthy control subjects were involved in this study. The visual perception of all eight patients was assessed with the motor-free visual perception test (MVPT), and their SLFs were reconstructed using DTI. The results showed that there was a significant difference between the DTI parameters of the patients and the control subjects. Moreover, patients with microstructural damage to the right SLF showed impairment of visual perception. In patients with damage to both the dorsal and ventral pathways of the right SLF, spatial neglect was present. However, although a leftward SLF asymmetry was revealed in our patients, this lateralization did not show a relationship with visual perception. In conclusion, the microstructural changes of the right SLF play an important role in visual perception, and both pathways contribute to spatial neglect, but leftward lateralization of the right SFL activity after a stroke does not contribute to general visual perception.

## 1. Introduction

Visual perceptual deficits are primarily the result of right cerebral hemisphere damage that occurs following a stroke [1]. These deficits have been explained as a deterioration in the process of unconscious inference from the information contained in the visual images produced by interactions from the frontal to the parietal lobe [2,3]. Impairments of visual perception are present in approximately 35 to 75% of patients with brain damage, and often lead to severe restrictions of independent functioning [3]. In these patients, the processing of visual information is often impaired. In some patients, an essential part of this visual information, usually on the left side, is neglected, which has been defined as spatial asymmetry when processing information after brain injury [4,5,6,7,8]. A comprehensive assessment of perceptual impairments can be conducted using the motor-free visual perception test (MVPT) [3,9].

The primary neural tract associated with visual perception is the superior longitudinal fasciculus (SLF) [10,11]. The SLF is a large bundle of fibers connecting the parietal, occipital, and temporal lobes and is associated with core cognitive processes, including attention, memory, emotions, and language [10]. In particular, the left SLF is associated with a person’s verbal working memory, whereas the right SLF seems to be responsible for visual-spatial functions and attention [11]. A previous study has suggested that injury of the right SLF contributes to spatial neglect. Therefore, the right SLF plays an important role in spatial neglect [12].

Diffusion tensor tractography (DTT), derived from diffusion tensor imaging (DTI), is a magnetic resonance imaging (MRI) technique employed to measure the diffusion of water molecules in neural fibers. It allows for the visualization and reconstruction of neural tracts, and may reveal injury of these tracts not detected by conventional MRI [4,6,13,14,15,16,17,18,19]. Many studies have reported that microstructural fiber changes after a stroke influences cognitive functions, and the lateralization index of the arcuate fasciculus, for example, has been shown to play an important role in the prognosis of language impairment after a stroke [4,13,20,21,22]. Several DTI studies have also investigated visual perception and spatial neglect [6,12]. However, to our knowledge, although DTI is useful to observe neural tract injury after a stroke, there have been no studies conducted to investigate the influence of various changes in the microstructure or the lateralization index of the SLF after brain injury on visual perception exploiting DTI. Therefore, the aim of this study is to investigate whether the various microstructural changes of SLFs after a stroke in the right hemisphere affect visual perception, including neglect.

## 2. Methods

### 2.1. Subjects

The patient group for this study consisted of eight patients (seven males and one female; mean age of 52.5 years; age range from 23 to 69 years) with brain damage in the right hemisphere (five patients with intracranial hemorrhage (ICH) and three patients with infarction). Sixteen subjects (four males and 12 females; mean age of 55.7 years; age range from 35 to 60 years) without a past medical history were included as the mean age-matched control group for this study (Table 1). The characteristics of the patients are summarized in Table 2. Treatments were as follows: patient 1 was a 68-year-old male with a history of hypertension who underwent a craniotomy for ICH; patient 2 was a 37-year-old male without a medical history who underwent a craniectomy and cranioplasty for ICH; patient 3 was a 23-year-old male without a past medical history who took cilostazol with atorvastatin; patient 4 was a 47-year-old male with a history of hypertension who underwent conservative treatment for ICH; patient 5 was a 61-year-old male with a history of hypertension and myocardial infarction who underwent conservative treatment for ICH; patient 6 was a 48-year-old male with no medical history who underwent a craniotomy of the right frontal area and hematoma evacuation; patient 7 was a 67-year-old male with no past medical history who underwent a thrombectomy at the right middle cerebral artery (MCA) occlusion and took aspirin for treatment with atorvastatin; patient 8 was a 69-year-old female with a history of arterial fibrillation who underwent a thrombectomy at the right MCA and was administrated micronized rivaroxaban with atorvastatin. The inclusion criteria were as follows: (1) first-ever stroke, (2) brain injury in the right hemisphere, and (3) no history of traumatic brain injury or a psychiatric disorder. Patients with contraindications for MRI, including the insertion of a pacemaker, moderate to severe cognitive impairment (the Korean Mini-Mental Status Examination (K-MMSE) < 14), or aphasia, were excluded [23,24] (Figure 1 and Table 2). This study included a retrospective protocol, was in accordance with the Declaration of Helsinki, and was approved by the Inha University Hospital Institutional Review Board (IRB No. 2020-04-025) on 20 April 2020.

### 2.2. Visual Perception Tests

All of the patients were assessed in terms of visual perception using the motor-free visual perception test (MVPT). On the same day, an MRI was performed. The MVPT was introduced by Colarusso and Hammill in 1972 [9]. It consists of 36 items and measures five dimensions of visual perception, namely, visual discrimination, figure-ground discrimination, spatial relationship, visual closure, and visual memory [25]. The parameters of the MVPT consist of a raw score, visual perception processing time, and left or right response (Table 1 and Appendix A). The score range of the left response behavior may vary from 0 to 21, and lower scores of the left response indicate severe neglect (normal values: in the age group 18–49 years, 18 or more, and in the age group 50–69 years, 18 or more). The raw score (the total score is 36) represents the severity of visual perception, and lower scores indicate visual perception impairment (normal values: in the age group 18–49 years, 34–36, and in the age group 50–69 years, 32–36). A slow visual perception time (normal values: in the age group 18–49 years, 2.5–4.0 s, in the age group 50–69 years, 3.0–5.4 s, and in the age group 70–80 years, 4.5–7.1 s) for each plate also indicates severe visuospatial perception impairment [9,25,26]. 

### 2.3. MRI Acquisition

The DTI images of all patients were acquired at approximately four or five weeks after onset using a SIGNA™ Architect 3.0T MRI system (General Electric, Milwaukee, WI, USA). The MRI conditions for the DTI parameters were as follows: 30 directions and 72 contiguous slices, with a slice thickness = 2 mm, field of view = 240 × 240 mm^2^, acquisition matrix of 128 × 128 matrix, b = 1000 mm^2^/s, repetition time (TR) = 15,000 ms, and echo time (TE) = 80.4 ms. An analysis of the DTI images was performed using DTI studio software (www.mristudio.org, Johns Hopkins Medical Institute, Baltimore, MD, USA). In five of the eight patients with ICH, T2-weighted gradient recalled echo (GRE) images were obtained to observe the blooming artifacts near the hemorrhages [27]. 

### 2.4. Fiber Tracking

We reconstructed the SLF using the superior (‘OR’ region of interest (ROI)) and inferior (‘AND’ ROI) portions of the SLF around the transverse temporal gyri, superior temporal gyri, and pars orbitalis [28]. The conditions of fiber tracking were as follows: fractional anisotropy (FA) of <0.15 and turning angle of >60° [29]. We measured the DTI parameters, including the fractional anisotropy (FA), tract volume (TV), and mean diffusivity (MD), as shown in Figure 2. To compare the SLF of patients with the control SLF, the SLF was divided into dorsal and ventral pathways [30].

### 2.5. Quantitative Analysis

The asymmetries of the reconstructed tract after a stroke were evaluated by calculating the lateralization index (LI) for TV [31]. We defined the lateralization index for TV after a stroke as the lateralization index for tract volume (LI-TV). The formula of LI-TV was LI-TV = (L − R)/(L + R), where L and R were the tract measurement for TV. The range of LI-TV was −1 to 1, and the index values were −0.1 ≤ LI-TV ≤ 0.1 (bilateral lateralization). LI-TV with a ≥0.1 value indicated left-hemispheric lateralization and ≤ −0.1 suggested rightward asymmetry [31,32]. 

### 2.6. Statistical Analysis

The quantitative data are expressed in terms of means ± standard deviations. The statistical differences between the DTI parameters or LI-TV of each patient and those of the control group were defined as a deviation from the reference values of at least two standard deviations (SDs) [14]. The Mann–Whitney U test was used to analyze the difference between the patient group and the control group.

## 3. Results

This study investigated the microstructural change of SLFs after a stroke in the right hemisphere using DTI and how this affected visual perception. Table 1 list the demographical characteristics of our eight patients with a stroke in the right hemisphere (five patients with hemorrhage and three patients with infarction). The range of the K-MMSE scores of the patients varied from 17 to 30. The raw scores of the MVPT in all of the patients, except for patient 3, were in the impaired range, and all of the patients, except patients 3 and 8, exhibited abnormally slow processing times. Patients 2, 4, 6, 7, and 8 had abnormal left response scores (Table 2 and Table 3). 

Before the reconstruction, no blooming artifacts were observed near the hemorrhage in the GRE images of all five patients with ICH. Although the right and left SLF tracts were reconstructed for all of the patients, the results of the reconstruction of the right SLF showed a structural change (discontinuation and narrowing) of this SLF in all of the patients compared with the control subjects. Patients 1 and 3 showed discontinuation in the ventral pathways. The right SLF in patients 2, 4, 6, 7, and 8 indicated discontinuation of the dorsal and ventral pathway in DTI. Although the ventral pathway of the right SLF in patient 5 was preserved compared with that of the control group, the dorsal pathway was damaged (Figure 3). 

We found a statistical difference in the FA, TV, and MD values of the right SLF in patients 1, 2, 4, 6, 7, and 8 compared with the control subjects. The TV and MD values of patient 5 revealed a significant difference compared with the control group. However, the DTI parameters of patient 3 did not significantly differ from the controls. The DTI parameters in the left SLF of all of the patients did not show a significant difference compared with the control subjects. The mean LI-TVs of the control subjects were −0.1 ≤ LI-TV ≤ 0.1, and this indicated bilateral lateralization. As the LI-TVs were ≥−0.1 in all of the patients with a stroke, their SLF indicated left hemispheric lateralization. Compared with the control subjects, the mean LI-TV of all of the patients was two SDs above the values of the control group (Table 4). Additionally, the FA, TV, and MD values, and the LI-TV of the right SLF between the patient and control groups, showed a statistical significance (*p* < 0.05; Table 4).

## 4. Discussion

This study showed that the microstructural change and lateralization of SLFs, reconstructed using DTI following a right hemisphere stroke, affected visual perception. All of the results are summarized in Table 5. The structure of the right SLF following a stroke in the right hemisphere was substantially changed compared with the control subjects. In most patients, the FA, TV, and MD values of the right SLF revealed a significant difference when compared to those of the control group of healthy subjects; the LIs were positive, indicating left hemisphere lateralization after a stroke. In patients with abnormal FA, TV, or MD values compared with the control group, an abnormal raw score was shown. In patients with discontinuation of both ventral and dorsal pathways of the right SLF, abnormality of the left response was shown. Patient 3, with normal values of DTI parameters, did not show abnormal MVPT results (Table 5). 

The FA values represent the directionality at a microscopic level, at the level of the microstructure of axons, myelin, and of microtubules [15,33,34]. The TV values indicate the number of voxels within the neural tracts [15]. Therefore, decrements in the values of FA and TV indicate neural tract injury [19]. The MD values showed the quantitation of water diffusion, and the differences reflected the variation within the inter- and extracellular space and a decrement of neuropil [14,15,35,36]. Increases in the MD value have been shown to contribute to atrophy or differences in tissue density [36]. Therefore, changes in these parameters reflect the presence of neural injury [15,35,37,38]. Additionally, LI indicates microstructural white matter asymmetry [31]. Previous studies have suggested that arcuate fasciculus differences between the left and right hemispheres in healthy subjects exhibited a trend toward leftward lateralization, and the lateralization of arcuate fasciculus after a stroke increased, indicating an important role of language function in the prognosis of recovery after a stroke [20,21,22]. Based on this evidence, because the arcuate fasciculus is a component of the SLF, we investigated the lateralization of SLF following a stroke in the right hemisphere [10]. Our results show that, as in previous studies, the mean LI of the control group points toward bilateral lateralization. However, we found a positively increased LI value in patients compared with the control subjects, indicating a leftward asymmetry of the right SLF due to brain damage in the right hemisphere. 

Visual processing in the brain is composed of a dorsal and ventral visual stream [30]. The ventral stream has been found to be related to recognition and discrimination, whereas the dorsal stream is usually associated with an analysis of spatial location [30]. We performed DTI on the right SLF, and the results showed discontinuation of the ventral pathway (in patients 1 and 3), dorsal pathway (in patient 5), or both pathways (in patients 2, 4, 6, 7, and 8). Patients with a discontinuation of both pathways of the right SLF exhibited an abnormality of the left response, raw score, and processing time in a visual perception test. Patients with a discontinuation of the dorsal pathway showed an abnormal raw score and processing time, regardless of the normal left response in the same test. In line with these results, we suggest that both the dorsal and ventral pathways of the right SLF are responsible for spatial neglect, which is not consistent with a previous study [30]. However, unlike this previous study, which argues that the dorsal pathway is associated with spatial function, we propose that it is the interconnection between the dorsal and ventral pathways that plays an important role in spatial neglect [30]. Therefore, a further investigation of our proposal concerning the function of each pathway is required. 

Our findings revealed that in the patients with a microstructural change of the right SLF (patients 1, 2, 4, 5, 6, 7, and 8), an abnormal raw MVPT score was shown, while all of the patients indicated abnormal LI values. The DTI parameters of the left SLF in all of the patients revealed normal values. This result suggests that microstructure damage of the right SLF induces visual perception impairment. Additionally, as mentioned above, the arcuate fasciculus depends on lateralization and has the characteristic of leftward asymmetry in the healthy group. The change of lateralization due to brain damage obviously affects the prognosis of language function [20,21,22]. However, in this study, although the leftward lateralization of the SLF due to stroke is shown in all of the patients involved in this study, patient 3 does not have visual perception impairment. Therefore, this result is not associated with visual perception. A limitation of our study was that, although it used reconstruction of the whole-brain SLF, the SLF was not separated into SLF I, SLF II, and SLF III and its functions, or the lateralization of each component [31,39]. A previous study also exhibited a correlation between behavior and anatomical lateralization using the lateralization index of the three SLF branches. In this study, SLF II indicated direct communication between the dorsal and ventral networks [39]. Considering this, our future plan is to investigate the function of the three SLF components using the relationship between visual perception and DTI parameters. Additionally, several studies have suggested that visual perception deficits are associated with inferior longitudinal fasciculus and inferior fronto-occipital fasciculus damage as well as injury to the SLF [4,40,41]. Here too, further study on the relationship between these tracts and visual perception is required. 

An important finding of this study is that that pathway, DTI parameters, and LI of the right SLF were not associated with processing time. This result is not consistent with a previous study indicating that the processing speed is associated with the right SLF [42]. The previous study suggested that processing speed was related to fronto-parietal integration [42]. Therefore, the variability of the lesions of the right SLF in patients requires further study, using various ROIs in the dorsal and ventral pathways. 

This study has revealed that perception is influenced by the microstructural change of the right SLF in patients with a stroke in the right hemisphere, and lateralization of the SLF after a stroke is not related to visual perception. The results illustrate that, as shown in a previous study, the right SLF plays an important role in visual perception, while the left SLF does not [11]. In particular, both the dorsal and ventral pathways of the right SLF are responsible for spatial neglect. To the best of our knowledge, this study is the first to report that while the injury of both dorsal and ventral pathways in the right SLF contributes to visual perception, processing time is not responsible for damage of the right SLF. Additionally, although lateralization of a neural tract in all patients after a stroke in the right hemisphere was shown, the leftward lateralization of the same neural tract was not related to visual perception. In addition to the limitations mentioned above, this study should be complemented by larger scale and long-term studies. Finally, because DTI interpretation is operator-dependent, such as using “OR” and “AND” ROIs for the reconstruction, this might have resulted in potential performance bias [15]. 

## 5. Conclusions

In conclusion, a previous study indicated that the anatomical structure of SLF plays an important role in visual perception and neglect [38]. Our findings are consistent with this and show that the microstructural change of the right SLF due to a stroke in the right hemisphere induces an impairment of visual perception. Particularly, damage to both the dorsal and ventral pathways of the right SLF plays an important role in spatial neglect. However, lateralization following brain injury in the right hemisphere was leftward asymmetric, and this did not contribute to visual perception. In other words, we suggest that visual perception is affected by the microstructural change of the right SLF after a stroke, regardless of lateralization (Figure 4). Additionally, DTI is useful for analyzing lateralization and microstructural changes after a stroke. 

## Figures and Tables

**Figure 1 diagnostics-10-00641-f001:**
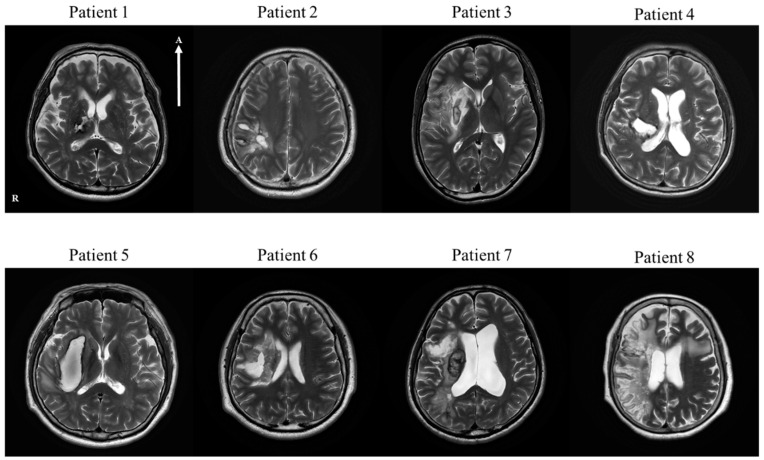
T2-weighted brain magnetic resonance images of all of the patients with stokes in the right hemisphere within four or five weeks after onset. In patients 1, 2, 4, 5, and 6, intracranial hemorrhage (ICH) in the right hemisphere is visible. Patients 3, 7, and 8 have a cerebral infarction in the right hemisphere. Patient 1: ICH in the right thalamus and posterior limb of the internal capsule. Patient 2: ICH in the right parietal lobe. Patient 3: cerebral infarction in the right frontal lobe, insular cortex, and basal ganglia. Patient 4: ICH in the right thalamus, posterior basal ganglia, and parietal lobe. Patient 5: ICH in the right basal ganglia. Patient 6: ICH in the right basal ganglia, right thalamus, right insula, and right front-parietal lobe. Patient 7: cerebral infarction in the right frontal lobe, insular cortex, and parietal lobe. Patient 8: cerebral infarction in the right basal ganglia and frontotemporal lobe. Note: ICH, intracranial hemorrhage; R, right; A, anterior. “R” and “A” in patient 1 are reflected throughout the patient’s image.

**Figure 2 diagnostics-10-00641-f002:**
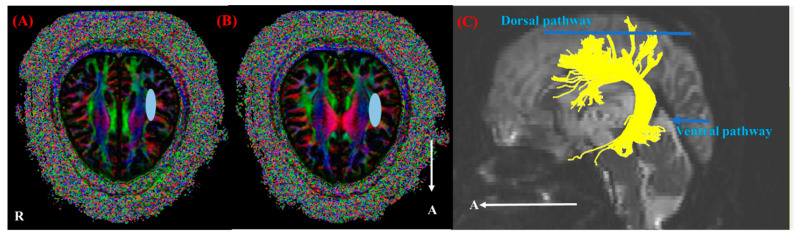
(**A**) “OR” region of interest (ROI) of the superior longitudinal fasciculus (SLF). (**B**) “AND” ROI of the SLF. (**C**) Reconstruction of the right SLF using diffusion tensor imaging (DTI) in a control subject (55 years, female). Note: yellow part; the right SLF, R, right; A, anterior; blue circle, ROI; blue line, dorsal pathway; blue arrow, ventral pathway.

**Figure 3 diagnostics-10-00641-f003:**
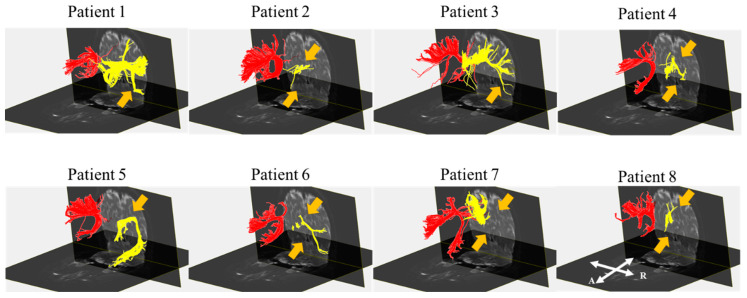
Diffusion tensor images for the bilateral SLFs of eight patients. Patients 2, 4, 6, 7, and 8 show an injury of both the dorsal and ventral pathways of the right SLF. Patients 1 and 3 show injuries in the ventral pathway of the SLF, and the SLF of patient 5 reveals damage in the dorsal pathway. Note: SLF, superior longitudinal fasciculus; yellow part; the right SLF, red part; the left SLF, R, right; A, anterior; orange arrow, injury site; “R” and “A” in patient 8 are reflected throughout the patient’s image.

**Figure 4 diagnostics-10-00641-f004:**
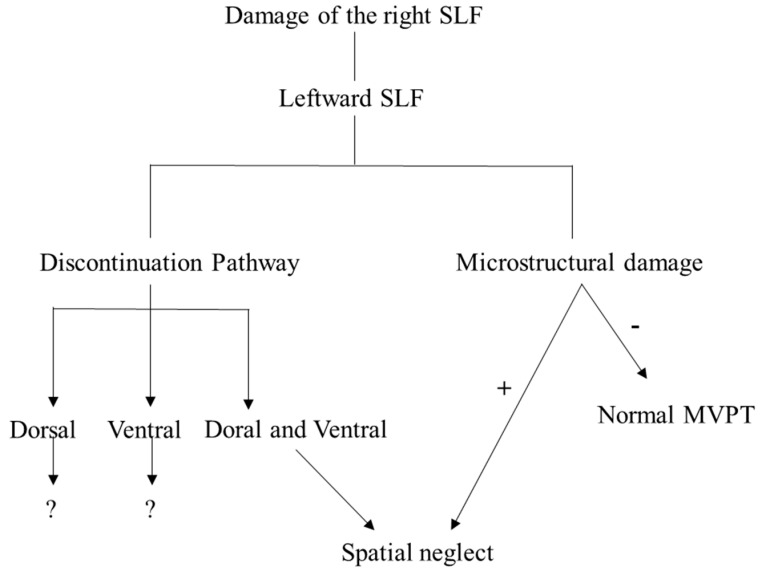
Summary of the pathway based on our finding. Note: SLF, superior longitudinal fasciculus. “?” means the requirement of further study. +; the right SLF with microstructural damage, −; the right SLF without microstructural damage.

**Table 1 diagnostics-10-00641-t001:** Parameters obtained for the present study.

	Patient Group (*n* = 8)	Control Group (*n* = 16)
Age (years)	52.50 ± 16.72	55.68 ± 6.61
Sex (M/F)	7/1	4/12
MMSE (scores)	24.88 ± 4.29	
MVPT		
Raw (scores)	20.63 ± 9.35	
Processing time (s)	10.28 ± 7.88	
Left response	13.13 ± 8.76	
DTI parameters of the right SLF		
FA	0.359 ± 0.091	0.504 ± 0.033
TV	2058.750 ± 1638.910	5203.88 ± 749.74
MD (×10^−3^ mm^2^/s)	0.854 ± 0.076	0.724 ± 0.301
LI-TV	0.515 ± 0.298	−0.007 ± 0.144

**Table 2 diagnostics-10-00641-t002:** Demographic data of eight patients and the results with the Korean Mini-Mental Status Examination (K-MMSE).

No	Age	Sex	Lesion	Type	MMSE	Medical History	Operation	Medication for Stroke
Patient 1	68	M	Thalamus and posterior limb of internal capsule, right	H	23	HTN	Craniotomy	
Patient 2	37	M	Parietal lobe, right	H	30		Craniectomy and Cranioplasty	
Patient 3	23	M	Frontal lobe, insular cortex and basal ganglia, right	I	28			cilotazol, atovastatin
Patient 4	47	M	Basal ganglia, frontoparietal lobe, right	H	17	HTN	Conservative treatment	
Patient 5	61	M	Basal ganglia, right	H	26	HTN, MI	Conservative treatment	
Patient 6	48	M	Basal ganglia, right	H	28		Craniotomy	
Patient 7	67	M	Frontal, parietal, temporal lobe and insula, right	I	26		Thrombectomy	aspirin, atorvastatin
Patient 8	69	F	Basal ganglia fronto-temporal lobe, right	I	21	AF	Thrombectomy	micronized rivaroxaban, atorvastatin

Note: M, male; F, female; H, hemorrhage; I, cerebral infarction; K-MMSE, Korean Mini-Mental Status Examination; HTN, hypertension; MI, myocardial infarction; AF, arterial fibrillation.

**Table 3 diagnostics-10-00641-t003:** Motor-free visual perception test (MVPT) results of eight patients.

		MVPT		
No	Age	Raw	Processing Time	Lt Response
Patient 1	68	18 *	14.27 *	21
Patient 2	37	28 *	4.89 *	16 *
Patient 3	23	35	3.16	21
Patient 4	47	25 *	8.56 *	16 *
Patient 5	61	26 *	6.84 *	20
Patient 6	48	14 *	27.00 *	11 *
Patient 7	67	11 *	13.23 *	0 *
Patient 8	69	8 *	4.27	0 *

Note: Lt, left; *, abnormal value.

**Table 4 diagnostics-10-00641-t004:** DTI parameters and lateralization index-tract volume (LI-TV) of SLFs of patients (*n* = 8) compared with the control group (*n* = 16; ^a^
*p* < 0.05).

No		Rt SLF			Lt SLF			LI-TV
		FA	TV	MD	FA	TV	MD	
Patient 1		0.417 *	3171.000 *	0.830 *	0.447	5970.000	0.783	0.306 *
Patient 2		0.337 *	610.000 *	0.926 *	0.492	7779.000	0.793	0.855 *
Patient 3		0.505	4472.000	0.741	0.495	6334.000	0.739	0.172 *
Patient 4		0.289 *	970.000 *	0.827 *	0.450	3831.000	0.765	0.596 *
Patient 5		0.446	3533.000 *	0.828 *	0.493	4642.000	0.745	0.136 *
Patient 6		0.326 *	652.000 *	0.837 *	0.484	4443.000	0.772	0.744 *
Patient 7		0.283 *	2127.000 *	0.982 *	0.479	6268.000	0.790	0.493 *
Patient 8		0.255 *	389.000 *	0.831 *	0.483	4438.000	0.769	0.839 *
	Avg	0.359	2058.750	0.854	0.478	5463.125	0.761	0.515
	SD	0.091	1638.910	0.076	0.018	1278.060	0.155	0.298
Control	Avg	0.504	5203.875	0.724	0.474	4996.625	0.749	−0.007
	SD	0.033	749.737	0.301	0.017	648.539	0.013	0.144
*p*-value		0.000 ^a^	0.000 ^a^	0.000 ^a^	0.697	0.490	0.106	0.000 ^a^

Note: SLF, superior longitudinal fasciculus; FA, fractional anisotropy; TV, tract volume; MD, mean diffusivity (×10^−3^ mm^2^/s); LI-TV, lateralization index-tract volume; DTI, diffusion tensor imaging; Avg, average; SD, standard deviation; Rt, right; Lt, left. * Parameters two standard deviations (SDs) above or below that of control subjects. ^a^ paraemters indicate statistical significance.

**Table 5 diagnostics-10-00641-t005:** A summary table of the results obtained for components of the MVPT and DTI parameters and LI-TV in the patient group (*n* = 8).

	MVPT			Discontinuation Pathway	DTI Parameter of Rt SLF	DTI Parameter of Lt SLF	LI
	Raw	Processing time	Lt response				
Patient 1	Abnormal	Abnormal	Normal	Vental	Abnormal	Normal	Abnormal
Patient 2	Abnormal	Abnormal	Abnormal	Ventral, Dorsal	Abnormal	Normal	Abnormal
Patient 3	Normal	Normal	Normal	Vental	Normal	Normal	Abnormal
Patient 4	Abnormal	Abnormal	Abnormal	Ventral, Dorsal	Abnormal	Normal	Abnormal
Patient 5	Abnormal	Abnormal	Normal	Dorsal	Abnormal	Normal	Abnormal
Patient 6	Abnormal	Abnormal	Abnormal	Ventral, Dorsal	Abnormal	Normal	Abnormal
Patient 7	Abnormal	Abnormal	Abnormal	Ventral, Dorsal	Abnormal	Normal	Abnormal
Patient 8	Abnormal	Normal	Abnormal	Ventral, Dorsal	Abnormal	Normal	Abnormal

Note: SLF, superior longitudinal fasciculus; LI, lateralization index for tract volume; DTI, diffusion tensor imaging; MVPT, motor-free visual perception test; Rt, right; Lt, left.

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
