# Peer review of "Relationship between Visual Perception and Microstructural Change of the Superior Longitudinal Fasciculus in Patients with Brain Injury in the Right Hemisphere: A Preliminary Diffusion Tensor Tractography Study"

_diagnostics, 2020, doi:10.3390/diagnostics10090641_

Round 1
Reviewer 1 Report
Kim et al. Manuscript entitled "Relationship between Visual Perception and Microstructural Change of the Superior –Longitudinal -Fasciculus in Patients with Brain Injury in the Right Hemisphere: A Preliminary Diffusion –Tensor -Tractography Study" is a very interesting article in which the authors investigated the associations of right hemisphere brain damage with respect to visual spatial deficits.
The strength of the article is: In this article the authors determined that any microstructural changes occurring at right SLF is central to regulate visual perception and spatial neglect, which is novel study. On the other hand, the authors found leftward laterization of the right SFL activity after stroke has no link in visual perception. However, the weakness of the article is below.
Line 11-12: What is unknown in the field? It will be better to add one more sentence.
Line 14-16: The style of the sentence for group comparisions should be more reflective. For example, you have mentioned one female, and I believe 7 are male. No transgender! These details can be added to the "Methods or Result section". If you found very noticeable changes in female then it is good to highlight in abstract section for each gender. Otherwise, patients (n=...) and p value could be better way of presentation.
Line 47: It is better if you open your sentence line like "In this study, we have exploited DTT.....".
Or, you may reiterate why did you choose DTT over other available technology (invasive or non-invasive)?
Line 60-81: This study was approved in April 20, 2020, thus you should mentioned whether patient has any exposure with other COVID-19 patient or high fever or other symptoms. Or, how the patients were handled during this pandemic in terms of safety? Example, wearing PPE, prescreening or symptom analysis for COVID-19 etc.
Line 60-65: There are many parentheses here, you should reiterate the group, sample size, sex, age, or other variables more clearly. If you highlight those variable here, you should also take into account in results section text for data.
Line 84: Since you have mentioned MVPT consists of 36-items. I would suggest to provide it (questionnaire or any documents to reflect it) in "Supplemental section".
Line 83-85: Here are many text that could be breakdown into simpler form by arranging those parameters in logical form together. You many arrange all those study parameters in the "box form" (Box 1) with appropriate groups titles (such as Age, gender, sample, size, standard value etc).
Line 96:Table:1: Please provide the appropriate font size or text, so that it is more readable.
Line 100-101: In the patient 1, there is text "R" or "A" inserted in the images . Please make sure it is reflecting throughout the patient's image but not just for upper panel figures.
I suggest to omit the figures from "methods section" and provide under "Result section"
There should be space here in the text "Figure1"
line 136-138: How you performed any statistical test to find p-value? If any, please mention here.
Line 139-140 : please arrange this figures in the "Result section"
Line 147: The opening sentences in the "result section" should briefly introduce the about the study or hypothesis or relevant text rather than jumping directly to explain Table 1.
Line 166-173: Please provide exact p-value to reflect significant difference and sample size. Please mention clearly whether you perform student's t-test, or ANOVA or other test.
Line 182-183: The font size, pattern should be consistent and readable. Currently, it is too small.
Line 186: In discussion section, it will be more prudent to priovide the sketch (Figure) about the summary of pathway based on your finding (flow chart or any other relevant sketch).
Author Response
Reviewer 1
Kim et al. Manuscript entitled "Relationship between Visual Perception and Microstructural Change of the Superior –Longitudinal -Fasciculus in Patients with Brain Injury in the Right Hemisphere: A Preliminary Diffusion –Tensor -Tractography Study" is a very interesting article in which the authors investigated the associations of right hemisphere brain damage with respect to visual spatial deficits.
The strength of the article is: In this article the authors determined that any microstructural changes occurring at right SLF is central to regulate visual perception and spatial neglect, which is novel study. On the other hand, the authors found leftward laterization of the right SFL activity after stroke has no link in visual perception. However, the weakness of the article is below.
- Line 11-12: What is unknown in the field? It will be better to add one more sentence.
Ans> Your opinion is right. Thank you
So, we add the sentence “Because various microstructural change of SLFs after a stroke in the right hemisphere affected visual perception, including negle, I”
- Line 14-16: The style of the sentence for group comparisions should be more reflective. For example, you have mentioned one female, and I believe 7 are male. No transgender! These details can be added to the "Methods or Result section". If you found very noticeable changes in female then it is good to highlight in abstract section for each gender. Otherwise, patients (n=...) and p value could be better way of presentation.
Ans> Your opinion is right. Thank you
In our group, there was no transgender and no noticeable change in female. So as your opinion, I deleted the “ one female”.
- Line 47: It is better if you open your sentence line like "In this study, we have exploited DTT.....".
Or, you may reiterate why did you choose DTT over other available technology (invasive or non-invasive)?
Ans> Your opinion is right. Thank you
Line 56-57 we add the sentence as your comments. “because DTI is useful to observe neural tract after a stroke, we have exploited DTI and”
- Line 60-81: This study was approved in April 20, 2020, thus you should mentioned whether patient has any exposure with other COVID-19 patient or high fever or other symptoms. Or, how the patients were handled during this pandemic in terms of safety? Example, wearing PPE, prescreening or symptom analysis for COVID-19 etc.
Ans> Your opinion is right. Thank you
Patients did not have any symptoms like high fever, cough, and rhinorrhea etc. the reason mentioned in manuscript is that date which patients underwent stroke was 2019. If you would like to mention any exposure, I will show the detail symptom.
- Line 60-65: There are many parentheses here, you should reiterate the group, sample size, sex, age, or other variables more clearly. If you highlight those variable here, you should also take into account in results section text for data.
Ans> Your opinion is right. Thank you
As your comment, there are many parentheses. So, we modified this paragraph. And we reiterate them below the sentence.
- Line 84: Since you have mentioned MVPT consists of 36-items. I would suggest to provide it (questionnaire or any documents to reflect it) in "Supplemental section".
Ans> thank you for your great comment
MVPT is performed using the below sheet. The sheet is simple because of using items like figure on book. So, I would like to have some question. Therefore, We attached supplement file as the below, but If you want to delete the file, We will remove that
- Line 83-85: Here are many text that could be breakdown into simpler form by arranging those parameters in logical form together. You many arrange all those study parameters in the "box form" (Box 1) with appropriate groups titles (such as Age, gender, sample, size, standard value etc).
Ans> Thank you for your great comment
We add the below box form.
Age (years) |
  |
Sex |
  |
MMSE (scores) |
  |
  |
  |
MVPT |
  |
Raw (scores) |
  |
Processing time (sec) |
  |
Left response |
  |
  |
  |
DTI parameters |
  |
FA |
  |
TV |
  |
MD (â…¹10-3mm2/s) |
  |
LI-TV |
  |
- Line 96:Table:1: Please provide the appropriate font size or text, so that it is more readable.
Ans> Your opinion is right. Thank you
We change the size to be more readable
- Line 100-101: In the patient 1, there is text "R" or "A" inserted in the images . Please make sure it is reflecting throughout the patient's image but not just for upper panel figures.
Ans> Your opinion is right. Thank you
So, we add the sentence” (“R” and “A” in the patient 1 is reflecting throughout the patient's image)
- I suggest to omit the figures from "methods section" and provide under "Result section"
Ans> thank you for your great comment
We provide figures under ”result section”
- There should be space here in the text "Figure1"
Ans> thank you for your great comment
We are space here in the text “Figure1”.
- line 136-138: How you performed any statistical test to find p-value? If any, please mention here.
Ans> thank you for your great comment
We performed statistical test “The statistical differences between the DTI parameters or LI-TV of each patient and those of the control group were defined as a deviation from reference values of at least two standard deviations (SDs)”
We performed this tool The reason to perform this tool is based on the results of many previous studies. The many previous studies were as the follows.
1) Neural Regen Res. 2013 Feb 15; 8(5): 474–478.
2) Neural Regen Res. 2017 Dec; 12(12): 2021–2024.
3) Front. Neurol., 02 August 2019 | https://doi.org/10.3389/fneur.2019.00787
4) Diagnostics 2020, 10(3), 156
- Line 139-140 : please arrange this figures in the "Result section"
Ans> thank you for your great comment
Figure 2 was arranged in the result section.
- Line 147: The opening sentences in the "result section" should briefly introduce the about the study or hypothesis or relevant text rather than jumping directly to explain Table 1.
Ans> thank you for your great comment
We add the opening sentences “The present study investigated microstructural change of SLFs after a stroke in the right hemisphere using DTI and this affected visual perception.”
- Line 166-173: Please provide exact p-value to reflect significant difference and sample size. Please mention clearly whether you perform student's t-test, or ANOVA or other test.
Ans> Your opinion is right. Thank you
Therefore, we add the statiscal tool
In Line 143, add “Mann-Whitney U test was used to analyze the difference between patient group and control group.”
- because we confirmed that our data is non-parametirc using Kolmogorov-Smirnov and Shapiro-Wilk
In line 201, add “ Additionally, FA, TV, MD values, and LI-TV of the right SLF between patient and control groups showed a statistical significance (P < 0.05, Table 2).”
- Line 182-183: The font size, pattern should be consistent and readable. Currently, it is too small.
Ans> Your opinion is right. Thank you
We change the size to be more readable
- Line 186: In discussion section, it will be more prudent to priovide the sketch (Figure) about the summary of pathway based on your finding (flow chart or any other relevant sketch).
Ans> Your opinion is right. Thank you for great comment
We add the figure based on your finding
Reviewer 2 Report
The study aims to find the possible correlation between visual perception and microstructural change in right brain hemisphere.
The authors use diffusion tensor imaging (DTI) to reconstruct the superior longitudinal fasciculus (SLF). They suggest that
the microstructural changes of the right SLF play an important role in visual perception. The study is within the scope of
the journal. But, I have several suggestion before further assessment. Please see my comments below:
You can make a table out of these histories:
"Patient 1 with a history of hypertension underwent a craniotomy 66 for ICH; patient 2 without a medical history
underwent a craniectomy and cranioplasty for ICH; 67 patient 3 without a past medical history took mono medication
(cilostazol) with atorvastatin; patient 68 4 with a history of hypertension underwent conservative treatment for
ICH; patient 5 with a history 69 of hypertension and myocardial infarction underwent conservative treatment for
an ICH ; patient 6 70 with no medical history underwent a craniotomy of the right frontal area and hematoma evacuation;
71 patient 7 with no past medical history underwent a thrombectomy at the right middle cerebral artery 72 (MCA)
occlusion and took mono medication for treatment (aspirin) with atorvastatin; and patient 8 73 with a history of
arterial fibrillation underwent a thrombectomy at the right MCA and was 74 administrated mono medication (micronized
rivaroxaban) with atorvastatin."
Did you perform statistically significant test or not for the data in Table 2?
"We found a statistical difference in the FA, TV, and MD values of the right SLF in patients 1, 2, 166 4, 6, 7, and 8 compared with control subjects."
Please show that in the table.
Why did you check two SDs difference?:
"Parameters two SDs above or below that of control subjects"
How accurate is the reconstruction algorithm:
"We reconstructed SLFs. The SLF was reconstructed using the superior (‘OR’ ROI) and inferior 122 (‘AND’ ROI) portions of the
SLF around the transverse temporal gyri, superior temporal gyri, and 123 pars orbitalis [28]."
Please add a table of acronyms as there are many abbreviations.
Please check Egnlish language once more. It is difficult to read some of the parts of the manuscript.
Author Response
Reviewer 2
The study aims to find the possible correlation between visual perception and microstructural change in right brain hemisphere. The authors use diffusion tensor imaging (DTI) to reconstruct the superior longitudinal fasciculus (SLF). They suggest that the microstructural changes of the right SLF play an important role in visual perception. The study is within the scope of the journal. But, I have several suggestion before further assessment. Please see my comments below:
- You can make a table out of these histories:
"Patient 1 with a history of hypertension underwent a craniotomy 66 for ICH; patient 2 without a medical history underwent a craniectomy and cranioplasty for ICH; 67 patient 3 without a past medical history took mono medication (cilostazol) with atorvastatin; patient 68 4 with a history of hypertension underwent conservative treatment for ICH; patient 5 with a history 69 of hypertension and myocardial infarction underwent conservative treatment for an ICH ; patient 6 70 with no medical history underwent a craniotomy of the right frontal area and hematoma evacuation; 71 patient 7 with no past medical history underwent a thrombectomy at the right middle cerebral artery 72 (MCA) occlusion and took mono medication for treatment (aspirin) with atorvastatin; and patient 8 73 with a history of arterial fibrillation underwent a thrombectomy at the right MCA and was 74 administrated mono medication (micronized rivaroxaban) with atorvastatin."
Ans> Thank you for great comment. Your opinion is right.
We split Table 1
Table 1; sex, age, MMSE, treatment
Table 2: MVPT result
- Did you perform statistically significant test or not for the data in Table 2?
"We found a statistical difference in the FA, TV, and MD values of the right SLF in patients 1, 2, 166 4, 6, 7, and 8 compared with control subjects."
Please show that in the table.
Ans> thank you for your great comment. Your comment is right
We suggested a definition of statistical difference “ The statistical differences between the DTI parameters or LI-TV of each patient and those of the control group were defined as a deviation from reference values of at least two standard deviations (SDs)” Line146
Also, we add the statistics between the patient group and healthy group using Mann-Whitney U test because we confirmed that our data is non-parametirc using Kolmogorov-Smirnov and Shapiro-Wilk
- Why did you check two SDs difference?:
"Parameters two SDs above or below that of control subjects"
Ans> thank you for your great comment
We performed statistical test “The statistical differences between the DTI parameters or LI-TV of each patient and those of the control group were defined as a deviation from reference values of at least two standard deviations (SDs)”
We performed this tool The reason to perform this tool is based on the results of many previous studies. The many previous studies were as the follows.
1) Neural Regen Res. 2013 Feb 15; 8(5): 474–478.
2) Neural Regen Res. 2017 Dec; 12(12): 2021–2024.
3) Front. Neurol., 02 August 2019 | https://doi.org/10.3389/fneur.2019.00787
4) Diagnostics 2020, 10(3), 156
- How accurate is the reconstruction algorithm:
"We reconstructed SLFs. The SLF was reconstructed using the superior (‘OR’ ROI) and inferior 122 (‘AND’ ROI) portions of the SLF around the transverse temporal gyri, superior temporal gyri, and 123 pars orbitalis [28]."
Ans> thank you for your great comment
Line 306 So, we suggested this problem in the limitation ,”because DTI interpretation is operator-dependent such as ‘OR; and ‘AND’ ROIs for the reconstruction”
- Please add a table of acronyms as there are many abbreviations.
Ans> thank you for your great comment
We add the abbreviations table
- Please check Egnlish language once more. It is difficult to read some of the parts of the manuscript.
Ans> thank you for your great comment
I edited English by native speaker in MDPI.
Round 2
Reviewer 2 Report
The manuscript has been improved substantially. I just have a few minor comments:
You can fill in the right column of Box 1. Maybe with averages or standard deviations.
Please cite Box 1 in the manuscript.
Author Response
You can fill in the right column of Box 1. Maybe with averages or standard deviations.
Please cite Box 1 in the manuscript
Ans>
Thank you for yor great comment
Please check the box 1
and we cite Box 1 in the manuscript Line 71 and 97